# Multi-Target Parameter Estimation of the FMCW-MIMO Radar Based on the Pseudo-Noise Resampling Method

**DOI:** 10.3390/s22249706

**Published:** 2022-12-11

**Authors:** Yao Jiang, Xiang Lan, Jinmei Shi, Zhiguang Han, Xianpeng Wang

**Affiliations:** 1State Key Laboratory of Marine Resource Utilization in South China Sea, School of Information and Communication Engineering, Hainan University, Haikou 570228, China; 2College of Information Engineering, Hainan Vocational University of Science and Technology, Haikou 571158, China

**Keywords:** target localization, FMCW-MIMO radar, DOA–range estimation, subspace leakage

## Abstract

Subspace methods are widely used in FMCW-MIMO radars for target parameter estimations. However, the performances of the existing algorithms degrade rapidly in non-ideal situations. For example, a small number of snapshots may result in the distortion of the covariance matrix estimation and a low signal-to-noise ratio (SNR) can lead to subspace leakage problems, which affects the parameter estimation accuracy. In this paper, a joint DOA–range estimation algorithm is proposed to solve the above issues. Firstly, the improved unitary root-MUSIC algorithm is applied to reduce the influence of non-ideal terms in building the covariance matrix. Subsequently, the least squares method is employed to process the data and obtain paired range estimation. However, in a small number of snapshots and low SNR scenarios, even if the impact of non-ideal terms is reduced, there will still be cases where the estimators sometimes deviate from the true target. The estimators that deviate greatly from targets are regarded as outliers. Therefore, threshold detection is applied to determine whether outliers exist. After that, a pseudo-noise resampling (PR) technology is proposed to form a new data observation matrix, which further alleviates the error of the estimators. The proposed method overcomes performance degradation in a small number of snapshots or low SNRs simultaneously. Theoretical analyses and simulation results demonstrate the effectiveness and superiority.

## 1. Introduction

As radar is widely used in autonomous driving and military fields, target localization has become one of the most important research directions. Although the traditional ultrasonic radar and laser radar [1,2,3] can achieve basic target detection, they are vulnerable to environmental effects. In harsh weather scenarios, the detection abilities of the above radars decrease sharply, including (but not limited to) the high costs and low estimation accuracies. To improve the detection performances in such complex communication environments, a FMCW-MIMO radar is proposed as a solution [4]. Compared to traditional radars, FMCW-MIMO radars have great advantages, including high accuracies [5,6], low interception probabilities [7], and strong anti-jamming abilities [8,9,10,11]. Meanwhile, they transmit orthogonal waveforms to form virtual array elements and enlarge the array apertures so that the limited array elements can be used to achieve higher degrees of freedom and spatial resolutions [12,13,14]. Thus, FMCW-MIMO radars have great potential for pedestrian detection or target localization [15,16,17], especially for multi-target scenarios.

Parameters of the FMCW-MIMO radar include the direction of arrival (DOA), the direction of departure (DOD), range, Doppler frequency (DF), etc. [18,19,20]. Recently, more focus has been placed on target localization work, which includes direction and range estimation. Fast Fourier transform (FFT) is proposed as a representative technology of the target localization [21]. In [22], a novel FFT algorithm for DOA estimation is proposed with the FMCW-MIMO radar. In the algorithm, the array aperture was expanded with interpolation technology. However, the estimation resolution was still limited by the number of sampling points. To further improve the resolution, a method was proposed in [23], which selects regions of interest in the total samples to reduce redundancy. While this method is FFT-based, the improvement in resolution is unsatisfactory compared to other algorithms. For the two-dimensional estimation problem, such as joint DOA–range estimation, the 2D-FFT [24,25] algorithm has high computational complexity, and the resolution is highly dependent on the number of sampling points.

Based on the disadvantages of the FFT algorithm, subspace-based algorithms have been studied, providing new perspectives for target localization. Parameter estimations based on the FMCW-MIMO system are more about extending the classical subspace-based algorithms into two-dimensional (2D) structures, such as the joint angle-frequency estimation (JAFE) [26,27] and 2D multi-signal classification (2D-MUSIC) algorithm [28]. In the 2D-MUSIC algorithm, the eigenvalue decomposition (EVD) [29] or singular value decomposition (SVD) [30] is employed to distinguish the signal and noise subspaces of the received covariance matrix. Based on the orthogonality between two subspaces, a two-dimensional spectral peak search function is applied to realize the parameter estimation through the joint search in the DOA–range domain [31]. However, the 2D-MUSIC algorithm requires decomposing the huge covariance matrix and selecting a suitable grid step to search for the two-dimensional spectral peaks, leading to great computation complexities and difficulties in real-time processing. To ensure the resolution and reduce the computational complexity simultaneously, Kim et al. attempted to combine the subspace algorithm with FFT [32,33]. First, the target parameters were roughly estimated by FFT. Then the MUSIC algorithm only searched for a small region that was close to the rough result by the FFT. By reducing the search region, the MUSIC algorithm avoids the redundant search computation. In [34,35], the authors reduced the computation complexity in a different way. The two papers respectively used ESPRIT and toot-MUSIC instead of MUSIC to estimate parameters, where the spectral peak search was replaced by the direct formula solution. To estimate the coherent source signals, a double smoothing algorithm was proposed in [36]. The algorithm constructs two double-smoothing matrices, and estimates the angle and delay, respectively, by using the translation invariance structure. However, the performances of the above subspace algorithms degrade rapidly in cases of small snapshot numbers or low SNRs. The reason is the subspace leakage [37,38,39], which means a part of the real signal subspace is possibly located in the estimated noise subspace.

In low SNRs or a small number of snapshot scenarios, the covariance matrices of sample data are different from theoretical covariance matrices. Diagonal loading [40] and shrinkage-based [41] methods move or scale eigenvalues to improve data covariance, but their signal and noise projection matrices do not change much. In [42], a random matrix theory and technology were introduced to improve the performance, but the performance was satisfactory only when the number of snapshots was considered.

In this paper, we propose a new joint DOA–range estimation algorithm for the FMCW-MIMO radar system to solve the performance degradation problem of the above algorithms under a small number of snapshots and low SNR. First, for angle estimation, we propose an improved algorithm based on the unitary root-MUSIC algorithm [43,44], which mainly optimizes and updates the covariance matrix of two-dimensional data to reduce the impacts of non-ideal items. Then, according to the results of the angle estimation, the least squares method is used to estimate the range information to realize the automatic matching of the angle and range. These angle and range estimators constitute the target estimation set. Finally, for the target estimation set, a threshold detection method based on PR technology is designed to detect and eliminate the abnormal estimation caused by the subspace leakage, which further improves the accuracy and stability of the target localization. The main contributions of the proposed method are summarized as follows:
The proposed method achieves a two-dimensional parameter estimation under the FMCW-MIMO radar system, which overcomes the disadvantages of traditional FFT-based and subspace-based algorithms. This algorithm has better performance in regard to a low SNR and a small number of snapshots.We propose a new iterative strategy to optimize the angle estimation and apply the least squares to the signal model of the FMCW-MIMO radar to achieve multi-target localization. Therefore, the range and angle parameters can be estimated in pairs.To further improve the target localization accuracy in case of a small number of snapshots and a low SNR, we introduce a threshold detection and pseudo-noise resampling algorithm in the joint estimation framework. Through the screening of the threshold detection and the improvement of the PR method, the unqualified estimates considered as outliers could be removed and the accuracy of the parameter estimation improved accordingly.

The rest of this article is organized as follows. Section 2 briefly introduces the signal model and parameters of the FMCW-MIMO radar. Section 3 describes the proposed algorithm in detail. In Section 4, numerical examples and simulation results are given and compared with other algorithms. Conclusions are drawn in Section 5.

The symbol definitions are shown in Table 1 to facilitate the derivation of subsequent formulas.

## 2. Signal Model

As shown in Figure 1, the MIMO radar uses two separate arrays to transmit and receive signals, respectively. The first row is the transmitting array, which contains two transmitting antennas; the receiving array lies on the last row, which consists of four antennas with equal spacings. Each pair of transmitting and receiving antennas can be independently equivalent to a monostatic element, which is marked as the midpoint of the transmit–receive element connection. In this way, a uniform linear array(ULA) composed of eight equivalent virtual elements is obtained, as shown in the middle row.

Figure 2 shows the working principle of the FMCW radar. When transmitting a frequency-modulated continuous signal, the transmitted signal can be specifically expressed as:(1)$$s\left(t\right)=V_{T}\cos\left(2\pi f_{c}t+\pi kt^{2}+\phi_{0}\right),0\leq t\leq T_{c}$$
where $V_{T}$ and $f_{c}$ denote the amplitude and starting frequency of the chirp, respectively. The slope of the chirp is given by $k=B/T_{c}$, which is related to the bandwidth *B* and the scan duration $T_{c}$. $\phi_{0}$ represents the initial phase of the signal and τ is the time delay between two chirps (TX and RX).

The target localization of the FMCW-MIMO radar is shown in Figure 3. The virtual array constructed by the MIMO part can be viewed as a ULA with spacing *d* between two adjacent elements. Signals are emitted from the array with the direction of departure θ. When reaching a target, echoes will be generated and impinge on the array from the direction of arrival θ. We assume that there are *K* far-field targets and *M* equivalent virtual antennas in the FMCW-MIMO radar system model.

According to the FMCW signal principle, the IF signal is generated after mixing, and the receiving signals are reflected by far-field fixed targets. After applying the Hilbert transform, the IF signal generated by the *k*-th target is:(2)$$u\left(t\right)=\mu e^{j(2\pi k\tau t+2\pi f_{c}\tau )}$$

As receiving elements receive, all IF signals generated by *K* targets, and the received signals of the *m*-th element can be expressed as:(3)$$x_{m}\left(t\right)=\sum_{k=1}^{K}\mu_{k}e^{j(2\pi k\tau_{m,k}t+2\pi f_{c}\tau_{m,k})}+n_{m}\left(t\right)$$
where $\tau_{m,k}=\frac{{2r}_{k}}{c}+\frac{2d\times \left(m-1\right)\sin\theta_{k}}{c}$ is the time delay of transmitting and th *m*-th receiving signals. $r_{k}$ is the range from the transmitting antenna to the *k*-th target, d×(m−1) denotes the range between the transmitting antenna and *m*-th receiving antenna, $n_{m}\left(t\right)$ is the Gaussian noise collected by the element.

N-point sampling is performed for Equation (Equation 3) with sampling interval $T_{S}$. The echo signal can finally be changed into:(4)$$\begin{array}{cc} x_{m}\left[n\right] & =\sum_{k=1}^{K}\mu_{k}e^{j2\pi k\left(\frac{2r_{k}}{c}+\frac{d\left(m-1\right)\sin\theta_{k}}{c}\right)nT_{S}}\\ \; & +e^{j2\pi f_{c}\left(\frac{2r_{k}}{c}+\frac{d\left(m-1\right)\sin\theta_{k}}{c}\right)}+n_{m}\left[n\right],n=0,1,2,\ldots ,N-1 \end{array}$$

Since the echo signals are from far-field, the spacings between the array antennas can be negligible compared to the distance between the transmitting antenna and the target, i.e., $r_{k}\gg d\times \left(m-1\right)\sin\theta_{k}$, Equation (Equation 4) can be reduced as: (5)$$\begin{array}{c} x_{m}\left[n\right]=\sum_{k=1}^{K}\mu_{k}e^{j(\frac{4\pi f_{c}d\left(m-1\right)\sin\theta_{k}}{c}+\frac{4\pi BT_{s}r_{k}n}{Tc}+\frac{4\pi f_{c}r_{k}}{c})}+n_{m}\left[n\right] \end{array}$$

The matrix form of the receiving signals can be shown:(6)$$\begin{array}{cc} X & =AS+N,X\in C^{M\times N} \end{array}$$

Here, the steering vector $a_{i}={\left[1,e^{j\frac{2\pi f_{c}d\sin\theta_{i}}{c}},\ldots ,e^{j\frac{2\pi f_{c}d\left(M-1\right)\sin\theta_{i}}{c}}\right]}^{T}$, i=1,2,…K constitutes the matrix $A=[a_{1},\ldots ,a_{k}]$. Moreover, the signal matrix $S={\left[s_{1},\ldots ,s_{K}\right]}^{T}$. The noise matrix N is a Gaussian white process with zero mean and covariance ${\sigma^{2}}_{n}I_{M}$.

## 3. Joint DOA–Range Estimation

This section proposes a novel method of joint DOA–range estimation in the FMCW-MIMO radar. First, for the complex-valued data processing, we introduce unitary transformations to convert complex-valued operations into real-valued operations. Then, in the angle domain, the covariance matrix is iteratively updated with the improved URM algorithm to extract more accurate DOA estimators. According to DOA estimators, we derive the corresponding steering vector and extract the phase. Range estimation is carried out with the least squares method, and the automatic pairing of two-dimensional parameters is directly realized. Finally, our work uses threshold detection for the paired parameter sets and optimizes the data matrix by pseudo-noise resampling. Each step of the proposed algorithm is elaborated in Figure 4.

### 3.1. Preliminary Estimation of DOA via URM Algorithm

Assuming that the noise and signals are independent of each other, the covariance matrix of the received data can be denoted:(7)$$R=E(XX^{H})=AR_{s}A+\sigma^{2}I_{M}$$
where $R_{s}$ means the covariance matrix of signals. In practical work, the theoretical value of $R_{s}$ is hardly obtained and an approach $\hat{R}$ by averaging *N* snapshots is used instead:(8)$$\hat{R}=\frac{1}{N}XX^{H}$$

Next, we decompose the covariance matrix $\hat{R}$ into signal and noise subspaces by the eigenvalue decomposition. For accelerating the decomposition, unitary transformation is used to change the covariance matrix $\hat{R}$ into the real-valued matrix $\hat{C}$:(9)$$\hat{C}=\frac{1}{2}Q_{M}^{H}\left(\hat{R}+J_{M}{\hat{R}}^{*}J_{M}\right)Q_{M}=Re\{Q_{M}^{H}\hat{R}Q_{M}\}$$
where $J_{M}$ is an M×M exchange matrix with ‘ones’ on its anti-diagonal and ‘zeros’; elsewhere. $Q_{M}$ is a sparse unitary matrix, defined as:(10)$$Q_{M}=\{\begin{array}{cc} \; & \frac{1}{\sqrt{2}}\left[\begin{array}{cc} I_{l} & jI_{l}\\ J_{l} & -jJ_{l} \end{array}\right],for\;M=2l\\ \; & \frac{1}{\sqrt{2}}\left[\begin{array}{ccc} I_{l} & 0_{l} & jI_{l}\\ 0_{l}^{T} & \sqrt{2} & 0_{l}^{T}\\ J_{l} & 0_{l} & -jJ_{l} \end{array}\right],for\;M=2l+1 \end{array}$$

Obviously, $Q_{M}$ is divided into two cases where the matrix dimension is odd or even. After the eigenvalue decomposition, $\hat{C}$ is expressed as follows:(11)$$\hat{C}=\hat{E}\hat{\Lambda }{\hat{E}}^{H}={\hat{E}}_{S}{\hat{\Lambda }}_{S}{\hat{E}}_{S}^{H}+{\hat{E}}_{N}{\hat{\Lambda }}_{N}{\hat{E}}_{N}^{H}$$
where ${\hat{E}}_{N}$ is the noise subspace. $\hat{\Lambda }=diag\left(\lambda_{1},\lambda_{2},\ldots ,\lambda_{M}\right)$ is the diagonal matrix composed of the eigenvalues λ. ${\hat{E}}_{N}$ consists of the eigenvector corresponding to the eigenvalue $\lambda_{K+1},\ldots ,\lambda_{M}$. Assuming that the targets are not correlated, the root polynomial is constructed according to the noise subspace [43,44].
(12)$$P_{URM}\left(z\right)=a^{T}\left(z^{-1}\right){\hat{E}}_{N}{\hat{E}}_{N}^{H}a\left(z\right)$$
where $a\left(z\right)={\left[1,z,...,z^{M-1}\right]}^{T},z=exp\left(jw\right)$. After selecting the *K* roots that are closest to the unit circle, the signal DOA can be calculated according to the following formula:(13)$$\hat{\theta }=\arcsin\left(\frac{\arg\left(z\right)\times c}{4\pi df_{c}}\right)$$

$\hat{\theta }$ denotes the angle parameters of the targets estimated by the unitary root-MUSIC (URM) algorithm.

### 3.2. DOA Estimation via Improved URM Algorithm

As mentioned above, the angle information required for multi-target localization can be gained by the URM algorithm. However, due to subspace leakage, the URM algorithm decreases in performance sharply with a low SNR or a small number of snapshots. The reason is the existence of non-ideal terms. If we expand the covariance matrix $\hat{R}$:(14)$$\begin{array}{cc} \hat{R} & =\frac{1}{N}\sum_{n=1}^{N}\{AS(n)+N(n)\}\{AS(n)+N(n){\}}^{H}\\ \; & =A\{\frac{1}{N}\sum_{n=1}^{N}S\left(n\right){S(n)}^{H})\}A^{H}+\frac{1}{N}\sum_{n=1}^{N}N\left(n\right){N(n)}^{H}\\ \; & +A\{\frac{1}{N}\sum_{n=1}^{N}S\left(n\right){N(n)}^{H})\}+\{\frac{1}{N}\sum_{n=1}^{N}N\left(n\right){S(n)}^{H})\}A^{H} \end{array}$$

In the preceding formula, $\hat{R}$ was divided into four terms, of which, the first two terms represent the signal covariance matrix and the noise covariance matrix, respectively. The focus is on the latter two items, which are defined as non-ideal terms. Assuming that the noise and signals are uncorrelated, when the number of snapshots is large enough, the non-ideal terms approach zero. However, in practice, the number of snapshots is limited, and the non-ideal terms adversely affect the algorithm, causing subspace estimation to deviate from the real subspace data.

In this section, we propose an improved URM algorithm to enhance the performance. The main idea of improvement is to revise the covariance matrix and reduce the impact of non-ideal terms through optimization iterations. Then, the angle parameters can be extracted from the modified ideal covariance matrix.

Firstly, the URM algorithm is used to estimate the targets and obtain the initial angle set ${\hat{\theta }}^{\left(0\right)}=\left[{\hat{\theta }}_{1}^{\left(0\right)},{\hat{\theta }}_{2}^{\left(0\right)},\ldots ,{\hat{\theta }}_{K}^{\left(0\right)}\right]$, then the updated array manifold is
(15)$$\hat{A}=\left[a\left({\hat{\theta }}_{1}^{\left(0\right)}\right),a\left({\hat{\theta }}_{2}^{\left(0\right)}\right),\ldots ,a\left({\hat{\theta }}_{K}^{\left(0\right)}\right)\right]$$

The superscript indicates the number of iterations. Accordingly, the signal $\hat{S}$ can be described as: (16)$${\hat{S}}^{\left(l\right)}=\arg\min_{a}{∥X-{\hat{A}}^{\left(l\right)}S∥}_{2}^{2}$$

After the least squares fitting, it can be expressed as:(17)$${\hat{S}}^{\left(l\right)}=\left[\left({\hat{A}}^{\left(l\right)}\right){}^{H}{\hat{A}}^{\left(l\right)}\right]{}^{\left((-1\right)}\left({\hat{A}}^{\left(l\right)}\right){}^{H}X$$

Therefore, the noise component can be estimated as:(18)$${\hat{N}}^{\left(l\right)}=X-{\hat{A}}^{\left(l\right)}{\hat{S}}^{\left(l\right)}$$

Introducing the real-valued covariance matrix $\hat{C}$ in Equation (Equation 11), the non-ideal term T can be found as:(19)$$\begin{array}{cc} T & ≜{\hat{A}}^{\left(l\right)}\{\frac{1}{N}\sum_{n=1}^{N}{\hat{S}}^{\left(l\right)}\left(n\right){{\hat{N}}^{\left(l\right)}\left(n\right)}^{H})\}\\ & ={\hat{A}}^{\left(l\right)}\{\frac{1}{N}\sum_{n=1}^{N}{\left(\left({\hat{A}}^{\left(l\right)}\right){}^{H}{\hat{A}}^{\left(l\right)}\right)}^{-1}{\left({\hat{A}}^{\left(l\right)}\right)}^{H}X\\ & \times (X^{H}-X^{H}{\left(\left({\hat{A}}^{\left(l\right)}\right){}^{H}{\hat{A}}^{\left(l\right)}\right)}^{-1}{\left({\hat{A}}^{\left(l\right)}\right)}^{H})\}\\ \; & ={\hat{P}}_{A}\{\frac{1}{N}\sum_{n=1}^{N}XX^{H}\left(I_{M}-{\hat{P}}_{A}\right)\}\\ \; & ={\hat{P}}_{A}\hat{C}{\hat{P}}_{A}^{⊥} \end{array}$$
where
(20)$${\hat{P}}_{A}≜{\hat{A}}^{\left(l\right)}{\left(\left({\hat{A}}^{\left(l\right)}\right){}^{H}{\hat{A}}^{\left(l\right)}\right)}^{-1}{\left({\hat{A}}^{\left(l\right)}\right)}^{H}$$
(21)$${\hat{P}}_{A}^{⊥}≜I_{M}-{\hat{P}}_{A}$$

As shown in the previous formula, the value of the non-ideal term is only related to the real-valued covariance matrix $\hat{C}$ and the array manifold after simplification.

On this basis, the covariance matrix is modified:(22)$${\hat{C}}^{\left(l\right)}={\hat{C}}^{\left(l-1\right)}-\epsilon (T+T^{H})$$

The parameters of the modified covariance matrix are estimated again to obtain the angle set $\left\{{\hat{\theta }}_{1}^{\left(l\right)},{\hat{\theta }}_{2}^{\left(l\right)},\ldots ,{\hat{\theta }}_{K}^{\left(l\right)}\right\}$. When the difference between two consecutive estimation results is less than the preset value, the iteration stops. The improved URM estimation method can be summarized as in Algorithm 1.
**Algorithm 1**: DOA estimation via the improved URM algorithm.**Input**: The received data ***X***1. The initial covariance matrix $\hat{C}$ is calculated according to (9).2. The initial angle set ${\hat{\theta }}^{\left(0\right)}=\left[{\hat{\theta }}_{1}^{\left(0\right)},{\hat{\theta }}_{2}^{\left(0\right)},\ldots ,{\hat{\theta }}_{K}^{\left(0\right)}\right]$ are calculated by (13).**Start iteration**3. Update the array manifold $\hat{A}$ according to Equation (Equation 15).4. Calculate the non-ideal terms by Equations (19) and (20).5. Update the covariance matrix $\hat{C}$ according to (22).6. Update the new angle set $\left\{{\hat{\theta }}_{1}^{\left(l\right)},{\hat{\theta }}_{2}^{\left(l\right)},\ldots ,{\hat{\theta }}_{K}^{\left(l\right)}\right\}$ according to the updated covariance matrix $\hat{C}$.7. $\sum_{k=1}^{K}{∥{\hat{\theta }}_{k}^{\left(l-1\right)}-{\hat{\theta }}_{k}^{\left(l\right)}∥}_{2}^{2}$ is less than the preset constant.**Terminate iteration****Output**: ${\hat{\theta }}^{\left(l\right)}=\left[{\hat{\theta }}_{1}^{\left(l\right)},{\hat{\theta }}_{2}^{\left(l\right)},...,{\hat{\theta }}_{K}^{\left(l\right)}\right]$

### 3.3. Range Estimation

After estimating the angle parameters, the steering vector matrix can be reconstructed as:(23)$$\tilde{A}=[a\left({\hat{\theta }}_{1}\right),a\left({\hat{\theta }}_{2}\right),\ldots ,a\left({\hat{\theta }}_{K}\right)]$$

Combined with Equation (Equation 9), we can obtain
(24)$$Y=\left({\tilde{A}}^{†}X\right){}^{T}=\left({\tilde{A}}^{†}\tilde{A}S+{\tilde{A}}^{†}N\right){}^{T}=S^{T}+\left({\tilde{A}}^{†}N\right){}^{T}$$

By normalizing the matrix, the real-valued vector matrix can be obtained:(25)Y(:,k)=Im{ln(Y(:,k)/Y(1,k))}

The ‘least squares’ is constructed as follows: (26)$$\min∥Gr_{k}-Y_{k}{∥}_{F}^{2}$$
where G is the least squares matrix: (27)$$G=\begin{array}{c} \begin{array}{c} \left[\begin{array}{cccc} 1 & 1 & ... & 1\\ 4\pi BT_{s}/Tc & 2\times 4\pi BT_{s}/Tc & ... & N\times 4\pi BT_{s}/Tc \end{array}\right] \end{array} \end{array}$$
where, $\Delta =4\pi BT_{s}/Tc$ denotes the phase difference between the adjacent elements and *N* is the number of snapshots. We define ${\hat{r}}_{k}$ as the estimator of the range parameter, which can be expressed as:(28)$${\hat{r}}_{k}=G_{r}^{†}Y\left(:,k\right)$$

The second element of each column vector of ${\hat{r}}_{k}$ is the estimated value of the range parameter. So far, in line with the proposed method, we can estimate the cursory DOA and range parameters of the objectives in pairs.

### 3.4. Refined DOA–Range Estimation by Pseudo-Noise Resampling

In Section 3.2 and Section 3.3, we propose a joint DOA–range estimation method based on the FMCW-MIMO radar. However, the abnormal estimators that deviate from real targets still occur in low SNR or a small number of snapshot scenarios. So a joint application of pseudo-noise resampling technology and threshold detection is proposed to mitigate the influence of outliers.

First, we use threshold detection to detect the outliers. The threshold range can be determined by conventional beamforming technology and FFT. Beamforming technology and FFT can quickly estimate the angle and range parameter at the expense of accuracy, to provide us with the approximate range of the targets. Thus, the thresholds for the angle and range parameters are assumed as:(29)$$\theta_{H}=\left[\theta_{1}^{max}-\theta_{1}^{L},\theta_{1}^{max}+\theta_{1}^{R}\right]\cup \left[\theta_{2}^{max}-\theta_{2}^{L},\theta_{2}^{max}+\theta_{2}^{R}\right]\cup \ldots \cup \left[\theta_{K}^{max}-\theta_{K}^{L},\theta_{K}^{max}+\theta_{K}^{R}\right]$$
(30)$$r_{H}=\left[r_{1}^{max}-r_{1}^{L},r_{1}^{max}+r_{1}^{R}\right]\cup \left[r_{2}^{max}-r_{2}^{L},r_{2}^{max}+r_{2}^{R}\right]\cup \ldots \cup \left[r_{K}^{max}-r_{K}^{L},r_{K}^{max}+r_{K}^{R}\right]$$
where $\theta_{max}$ and $r_{max}$ denote the K highest peak of the beamformer and FFT method, respectively, $\theta_{K}^{L}$ and $r_{K}^{L}$ represent the left boundary of the angle parameter and range parameter of each target interval, $\theta_{K}^{R}$ and $r_{K}^{R}$ are the right boundaries.

We denote the rough parameter estimation results with the set $\hat{\theta }=\left[{\hat{\theta }}_{1},{\hat{\theta }}_{2},\ldots ,{\hat{\theta }}_{K}\right]$ and $\hat{r}=\left[{\hat{r}}_{1},{\hat{r}}_{2},\ldots ,{\hat{r}}_{K}\right]$ in Section 3.2 and Section 3.3. If the estimated values satisfy the threshold detection, then the estimated parameters are taken as the final results. Estimators that do not pass the threshold detection are considered outliers, a pseudo-noise approach is presented when an outlier occurs. The idea of this approach is to perturb the raw measurement data matrix X by an artificially generated pseudo-random noise. Here, we define the pseudo-noise matrix $Y\in C^{M\times N}$ as the pseudo-noise matrix, where:(31)$$E\left(Y\right)=0,E\left(YY^{H}\right)=\sigma_{Y}^{2}NI,E\left(YY^{T}\right)=0$$

Meanwhile, as the variance is related to the variance of the original data matrix, the power of pseudo-random noise $\sigma_{Y}^{2}$ can be estimated by $\sigma_{Y}^{2}=\rho \sigma^{2}$, where ρ is a user-defined parameter and $\sigma^{2}$ is given by:(32)$$\sigma^{2}=\frac{1}{M-K}Tr{\hat{\Lambda }}_{N}=\frac{1}{M-K}\sum_{i=K+1}^{M}{\hat{\lambda }}_{i}$$

${\hat{\lambda }}_{i}$ is a group of eigenvalues arranged in descending order, which is achieved by the decomposition of matrix X.

Combing Equations (31) and (32), the resampled data matrix Z is:(33)Z=X+Y

We assume that the original matrix X is perturbed for W times and the W groups of resampled data matrix Z are obtained correspondingly. For each resampled data matrix, it is necessary to perform the steps in Section 3.2 and Section 3.3. The corresponding estimators construct the W groups of the estimators set $({\hat{\theta }}_{k},{\hat{r}}_{k}),k=1,2,\ldots ,K$. Then the threshold detection is applied to the estimator sets, and the W groups of the estimators are divided into two subsets:(34)$$\begin{array}{cc} \; & B_{1}=\{{\left({\hat{\theta }}_{k},{\hat{r}}_{k}\right)}^{\left(1\right)},\ldots ,{\left({\hat{\theta }}_{k},{\hat{r}}_{k}\right)}^{\left(J\right)}\},\\ \; & B_{2}=\{{\left({\hat{\theta }}_{k},{\hat{r}}_{k}\right)}^{\left(1\right)},\ldots ,{\left({\hat{\theta }}_{k},{\hat{r}}_{k}\right)}^{\left(W-J\right)}\} \end{array}$$

*J* groups of the estimator sets that pass the detection constitute $B_{1}$, and $B_{2}$ is composed of W−J groups of estimator sets rejected by the threshold detection.

If 0<J<W, the final target parameter estimator is the average result of the successful *J* groups of the estimator set:(35)$${\hat{\theta }}_{k}=\frac{1}{J}\sum_{i=1}^{J}{\hat{\theta }}_{k}^{\left(i\right)},{\hat{r}}_{k}=\frac{1}{J}\sum_{i=1}^{J}{\hat{r}}_{k}^{\left(i\right)}$$

If J=0 means that all groups of estimators fail to pass the threshold detection even if the PR technology is performed. Facing the case where all estimators deviate from the true value, we need to select the optimal one among the multiple groups of estimators set in $B_{2}$ as the final output estimation. Since the angle and range parameters of the target exist in pairs in the set, we can use one of the parameter information pieces as a judgment object. In this paper, the angle parameter was selected as the judgment object.

First, the angle estimators in $B_{2}$ are divided into *K* subsets according to different targets:(36)$$B_{2k}=\{{\hat{\theta }}_{k}^{\left(1\right)},\ldots ,{\hat{\theta }}_{k}^{\left(W\right)}\},i=1,\ldots ,K$$
where $B_{2k}$ represents the set of W group angle estimators for the k-th target.

According to Equation (Equation 13), the angle estimation is related to the corresponding polynomial root.
(37)$$\left|D_{k}\right|=\{\left|{\hat{z}}_{k}^{\left(1\right)}\right|,\ldots ,\left|{\hat{z}}_{k}^{\left(W\right)}\right|\},k=1,\ldots ,K$$

In the above DOA algorithm, the root polynomial $P_{URM}$ has (M-1) pairs of symmetrically mirrored roots about the unit circle, and the *K* roots closest to the unit circle are what we need. So, we choose the $\hat{z}$ with the smallest modulus in $D_{k}$ to ensure that the deviation between the corresponding estimated value ${\hat{\theta }}_{k}$ in $B_{2k}$ and the theoretical value is minimum. Then the paired range parameters will be determined. Finally, the corresponding target parameters $\left({\hat{\theta }}_{k},{\hat{r}}_{k}\right)$ can be obtained. The above steps can be summarized as Algorithm 2.
**Algorithm 2**: Refined DOA–range estimation by pseudo-noise resampling.**Input**: The received data ***X***, initial parameter estimator $\left({\hat{\theta }}_{k},{\hat{r}}_{k}\right)$.1. Set the threshold area and apply the threshold detection to these estimators.2. If it passes the detection, terminate this algorithm and output the estimator of $\hat{\theta }$ and $\hat{r}$.3. If not, W groups of the resampled data are obtained by using the pseudo-noise matrix.   (1) For the updated X, use the algorithms in Section 3.2 and Section 3.3 to estimate the DOA and range.   (2) Perform the threshold detection again for W group estimators and categorize them as (34).      If J > 0, then estimate target parameters via (35).      If J = 0, then select target parameters by (36) and (37).**Output**:$\left\{{\hat{\theta }}_{k},{\hat{r}}_{k}\right\}$

## 4. Simulations

In this section, the proposed algorithm is compared with conventional algorithms. The parameter settings of the FMCW-MIMO radar are shown in Table 2.

In this experiment, three different targets are set at $\left(\theta_{1},r_{1}\right)=[-8.78^{\circ },10.73$ m], $\left(\theta_{2},r_{2}\right)=[2.43^{\circ },32.78$ m], and $\left(\theta_{3},r_{3}\right)=[12.07^{\circ },23.66$ m], respectively. The number of Monte Carlo trials is MC = 500.

### 4.1. 2D Point Cloud of the Target

First, to verify the localization performance, the proposed algorithm was simulated with SNR = 10, N = 50, where N is the number of snapshots. The results are shown in Figure 1 and the X and Y axes represent the angle and range parameters, respectively.

In Figure 5, the three target points can be distinguished, and the estimated targets are very close to the preset ones, which proves the reliability of our proposed algorithm.

### 4.2. Performance Analysis

In this part, we set multiple SNRs and the number of snapshots to explore the differences between the proposed algorithm and the comparison algorithms in diverse environments. The 2D-MUSIC, 2D-ESPRIT, and DFT-root-MUSIC (DFT-RM) algorithms are used as comparison algorithms, among which 2D-MUSIC and 2D-ESPRIT are two-dimensional joint DOA–range estimation algorithms. Meanwhile, DFT-RM estimates the angle and range with the RM algorithm and DFT algorithm, respectively.

#### 4.2.1. RMSE Performance of DOA

The *RMSE* (root mean square error) is usually used as an important index to evaluate the algorithm’s performance. The *RMSE* of the FMCW MIMO radar’s DOA can be defined as:(38)$$RMSE_{\theta }=\frac{1}{K}\sum_{k=1}^{K}\sqrt{\frac{1}{MC}\sum_{mc=1}^{MC}{\left(\theta_{mc,k}-\theta_{k}\right)}^{2}}$$
where *K* denotes the number of targets, $\theta_{mc,k}$ stands for the angle estimation of the *k*-th target in the *mc*-th Monte Carlo experiment.

Figure 6 shows the *RMSE*s of the DOA estimation against SNR when the number of snapshots is fixed at 50. The results show that the proposed method is superior to other algorithms in accuracy and stability. The main reason is that the proposed method reduces the influences of non-ideal terms, which improves the accuracy of the location. The 2D-ESPRIT algorithm loses the array aperture when decomposing the subspace, so the *RMSE* is larger than that of the 2D-RM and proposed algorithm.

The trends of the *RMSE* with different snapshot numbers are revealed in Figure 7. In this simulation, the SNR is fixed at 5 dB, and the number of snapshots increases from 10 to 150. When the number of snapshots is less than 70, the performance advantage of the proposed algorithm is obvious, and even if the number of snapshots is greater than 70, it is also one of the algorithms with the smallest errors. When the number of snapshots exceeds 70, the proposed algorithm has a similar performance compared to 2D-RM and DFT-RM. As the number of snapshots increases, the deviation of the sample covariance matrix from the theoretical covariance matrix gradually shrinks. Therefore, the proposed method improves in an unobvious manner and gradually approaches 2D-RM with a large number of snapshots.

The CRB curve of the FMCW-MIMO radar is also given to evaluate the estimation performance of the algorithms. It shows that the proposed algorithm is closer to the CRB curve than the comparison algorithms in terms of the DOA estimation, which means that the proposed algorithm outperforms other algorithms, especially with low SNRs and a small number of snapshots.

#### 4.2.2. RMSE Performance of Range

In this simulation, we evaluated the range parameter estimation performances of different algorithms versus different SNRs and snapshots, as shown in Figure 8 and Figure 9. Similar to the previous section, the *RMSE* formula for the range of the FMCW MIMO radar can be written as:(39)$$RMSE_{r}=\frac{1}{K}\sum_{k=1}^{K}\sqrt{\frac{1}{MC}\sum_{mc=1}^{MC}{\left(r_{mc,k}-r_{k}\right)}^{2}}$$

Evaluating the range estimation is the same as the DOA estimation performance evaluation, with two different groups of parameter settings: SNR varies from −5 to 20 dB with N = 50, and the number of snapshots changes from 10 to 150 with SNR = 5 dB.

An explanatory point is needed for this simulation; DFT-RM uses the FFT algorithm for the range estimation, which highly depends on the number of snapshots. When the snapshot number is not enough, the range resolution will decrease, resulting in low accuracy and failure of the FFT algorithm.

Therefore, in Figure 8, when the number of snapshots is fixed at 50, the range estimation of the DFT-RM tends to be a straight line. In Figure 9, since the number of snapshots is always set in a low state, the DFT-RM performance is always unstable, resulting in a large estimation error. The performances of other algorithms all improve with the increase of the SNR and the number of snapshots, but the proposed one is closer to the CRB curve intuitively, as seen in Figure 8 and Figure 9. Therefore, the proposed method still has obvious advantages over the other conventional algorithms in the range domain.

#### 4.2.3. PSD versus SNR

The probability of successful detection (*PSD*) is another indicator to measure the performance, which can be defined by the following formula:(40)$$PSD=\frac{H}{D}\times 100$$
where *D* represents the number of experiments and *H* denotes the number of successful tests.

In this section, to demonstrate the trend of each algorithm completely, DOA and range estimates are considered successful in meeting $|\theta_{mc,k}-\theta_{k}|\;\leq 0.7^{\circ }$ and $|r_{mc,k}-r_{k}|\;\leq 1.5$ m.

Figure 10 and Figure 11 describe the *PSD* of the range and angle estimation under different SNRs when the number of N = 50. It is clear that the algorithm we proposed is better than other methods under different SNRs, especially in low SNRs. The reason is that the proposed algorithm reduces the negative impacts of non-ideal items. The algorithms, which are highly dependent on the sampling points or snapshots, suffer from non-ideal items and have more failed estimators.

#### 4.2.4. PSD versus Snapshots

In this section, we investigate the PSD of the proposed and compared algorithms with fixed SNRs and varying numbers of snapshots. In Figure 12, the proposed algorithm has a high success rate when the number of snapshots is only 30, and achieves full detection when the number of snapshots is 50. Compared with the RM, FFT-RM, and ESPRIT algorithms, the proposed method is more stable in the angle domain.

For the range estimation, Figure 13 shows the PSD curves of all algorithms from 10 to 150 snapshots. Compared to other algorithms, the proposed method has a higher success rate in all scenarios. Notice that the DFT-RM method has a low success rate level as this method requires more snapshots than others.

## 5. Conclusions

In this paper, the algorithm applies iterative updating to reduce the non-ideal items and approach a more accurate covariance matrix. Based on the covariance matrix, a rough angle estimation and range estimation were obtained by U-root-MUSIC and the least squares algorithms, respectively. Threshold detection was then used to refine the rough results, where the outliers were further processed. Simulations have shown that this method improves the estimation accuracy effectively, especially in conditions with a small number of snapshots and low SNRs. In future work, we will optimize the proposed algorithm to make it suitable for more practical application scenarios.

## Figures and Tables

**Figure 1 sensors-22-09706-f001:**
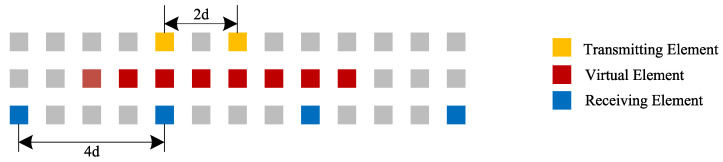
Synthetization of the virtual antenna array.

**Figure 2 sensors-22-09706-f002:**
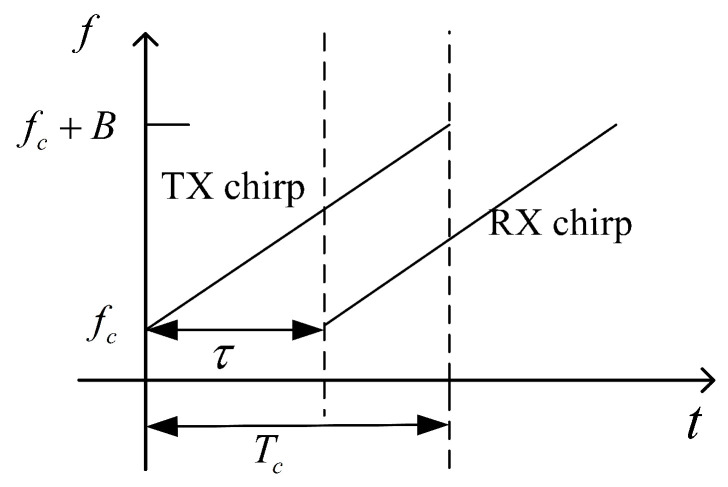
FMCW radar echo diagram.

**Figure 3 sensors-22-09706-f003:**
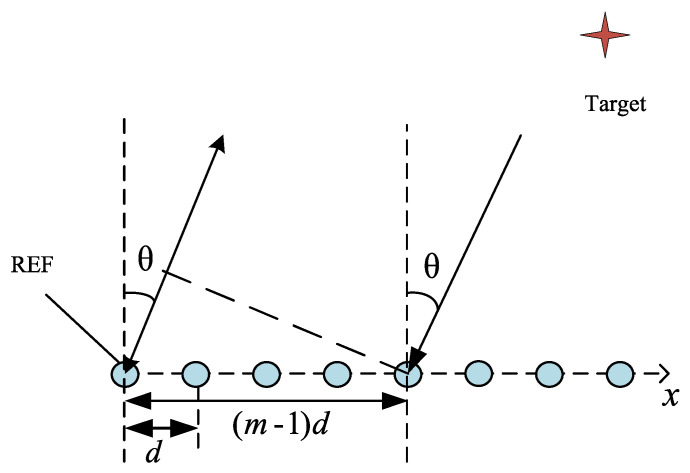
Target localization diagram of FMCW-MIMO radar.

**Figure 4 sensors-22-09706-f004:**
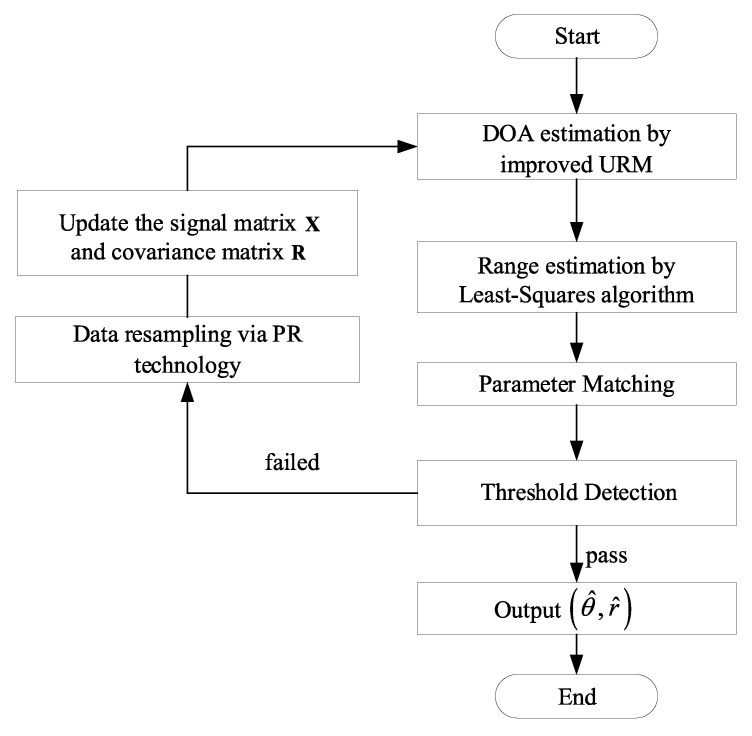
Structure of the proposed algorithm.

**Figure 5 sensors-22-09706-f005:**
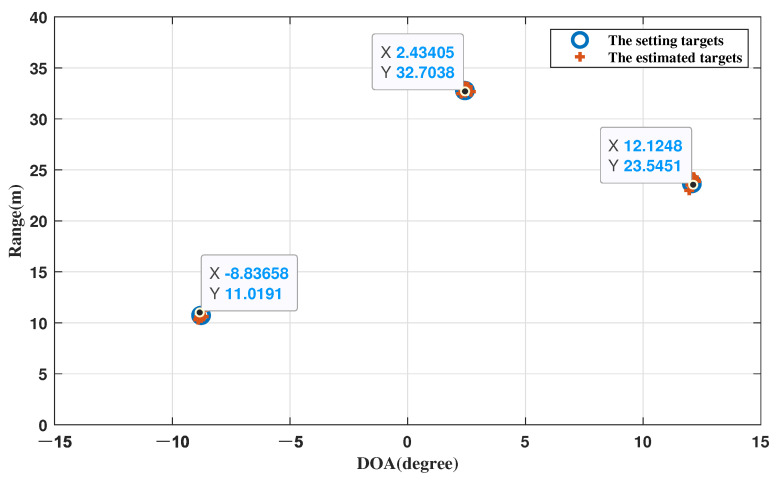
The 2D point cloud of the estimated targets.

**Figure 6 sensors-22-09706-f006:**
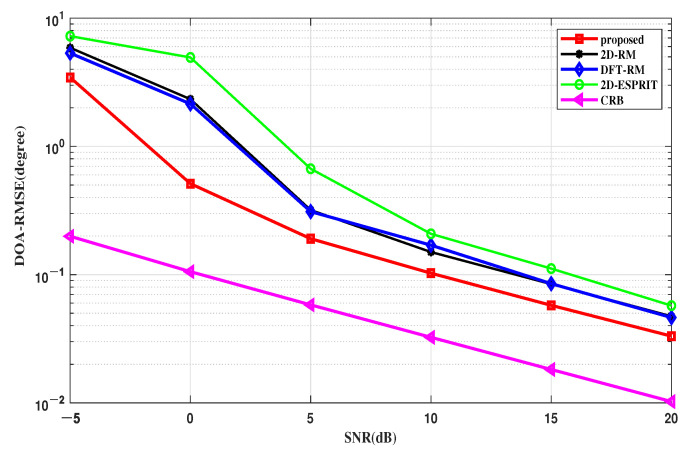
RMSE of the DOA estimation versus SNR.

**Figure 7 sensors-22-09706-f007:**
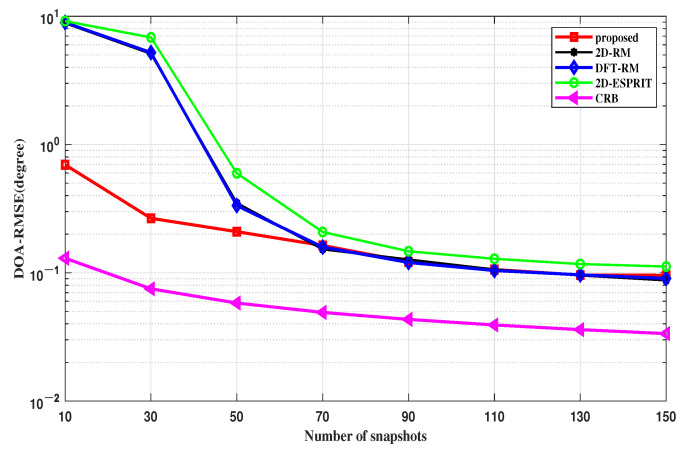
RMSE of the DOA estimation versus snapshots.

**Figure 8 sensors-22-09706-f008:**
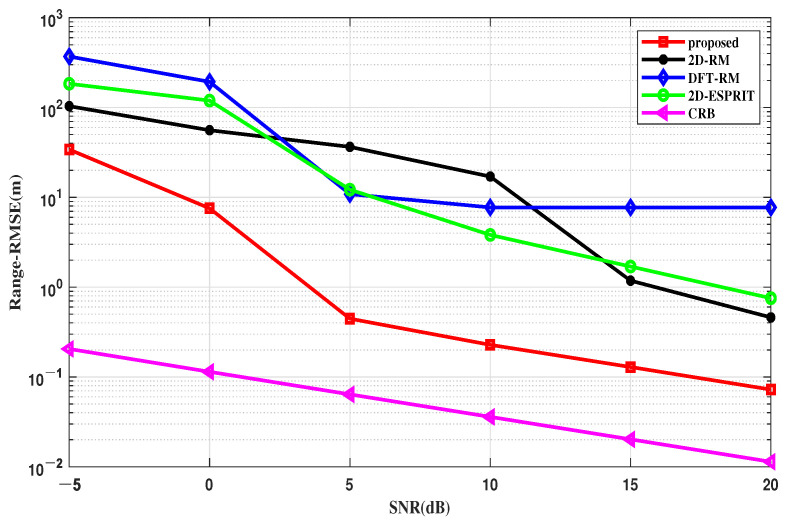
RMSE of the range estimation versus the SNR.

**Figure 9 sensors-22-09706-f009:**
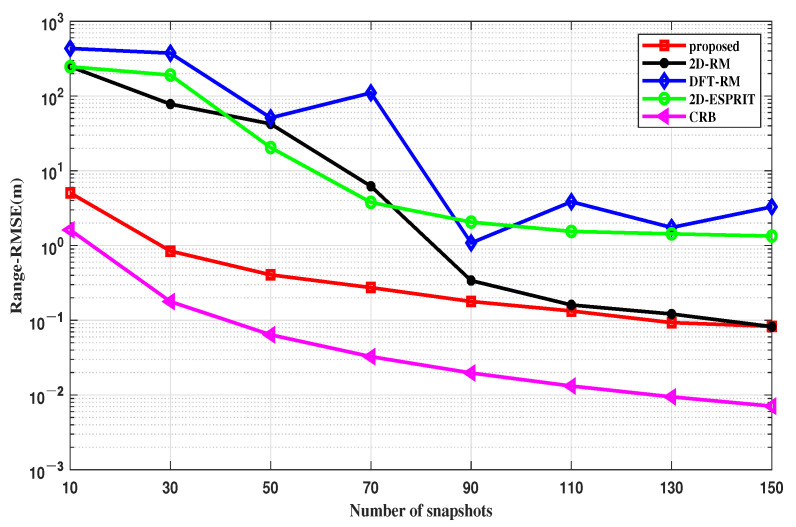
RMSE of the range estimation versus the snapshots.

**Figure 10 sensors-22-09706-f010:**
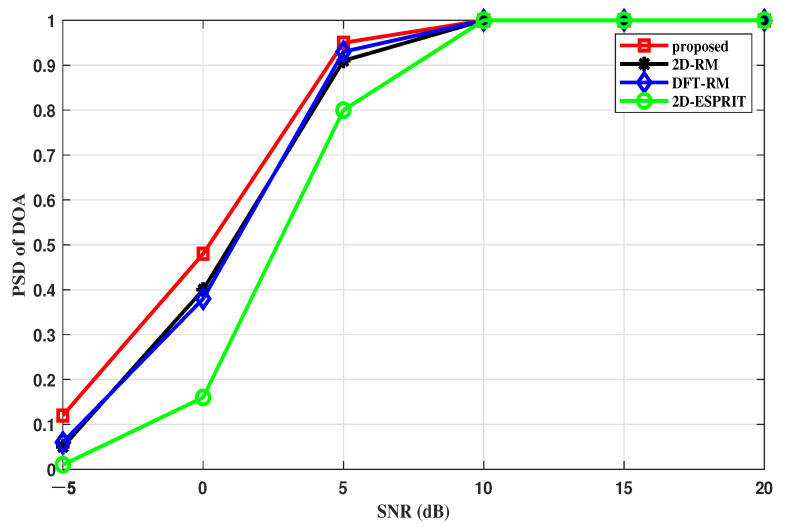
PSD of the DOA estimation versus SNR.

**Figure 11 sensors-22-09706-f011:**
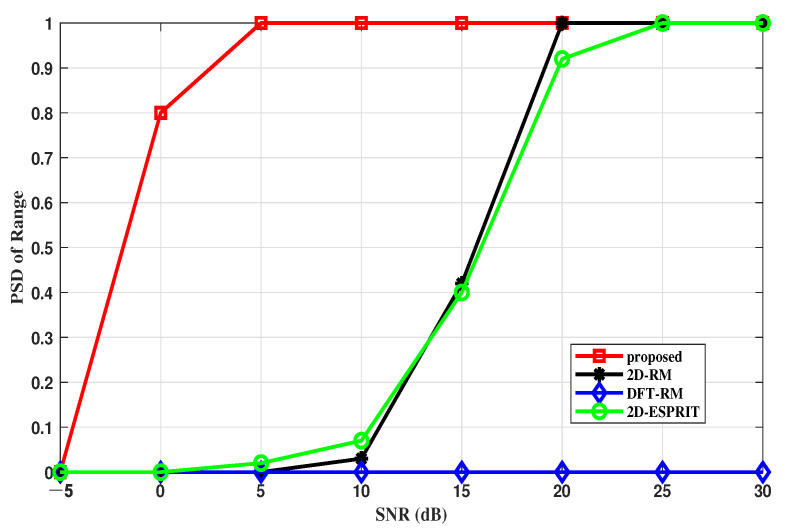
PSD of the range estimation versus SNR.

**Figure 12 sensors-22-09706-f012:**
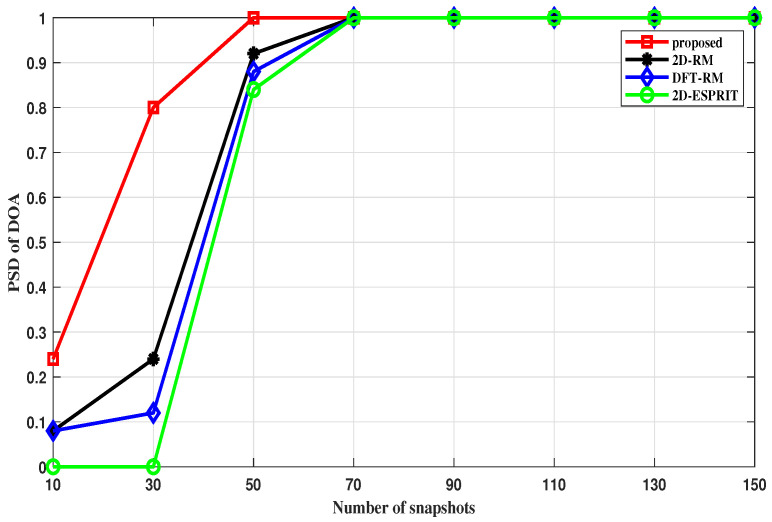
PSD of the DOA estimation versus the snapshots.

**Figure 13 sensors-22-09706-f013:**
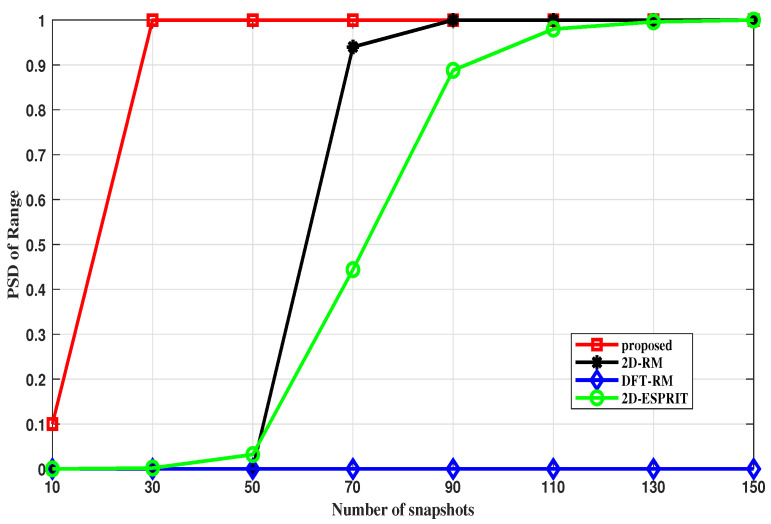
PSD of the range estimation versus the snapshots.

**Table 1 sensors-22-09706-t001:** Related notation.

Notations	Definitions
uppercase bold italic letter	matrix
lowercase bold italic letter	vector
$I_{M}$	identity matrix of *M* order
E(·)	mathematical expectation
diag(·)	diagonalization of the matrix
${\left(\cdot \right)}^{H}$	conjugate transpose of the matrix
${\left(\cdot \right)}^{T}$	transpose of the matrix
${\left(\cdot \right)}^{*}$	conjugate matrix
${\left(\cdot \right)}^{†}$	pseudo-inverse of the matrix
${\left(\cdot \right)}^{⊥}$	orthogonal complement of the matrix
$C^{M\times N}$	complex space of size M×N
$\left|\cdot \right|$	modulus operator
${∥\cdot ∥}_{2}$	$l_{2}$ norm operation
∪	union

**Table 2 sensors-22-09706-t002:** Parameters of the FMCW-MIMO radar.

Parameter	Value	Parameter	Value
c	3 × 10^8^ m/s	$T_{s}$	25 ns
$f_{s}$	2.46 × 10^9^ Hz	B	1.2 × 10^8^ Hz
λ	120 mm	M	8
d	60 mm	L	3

## Data Availability

Not applicable.

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
