# Peer review of "Multi-Target Parameter Estimation of the FMCW-MIMO Radar Based on the Pseudo-Noise Resampling Method"

_sensors, 2022, doi:10.3390/s22249706_

Round 1

Reviewer 1 Report

The authors are presenting a new approach for range and angle estimation, however, the paper is written with very low quality language (need a complete english overhaul) and the mathematical approach they take lacks proper definition and clearer concise derivation steps.

Some of my comments are:

1. some major inaccuracies starts in the introduction, the authors distinguish between MIMO radar and FMCW radar, as two separate 'technologies', where in fact, FMCW with MIMO array has been in the market for years. Also they claim, in lines 28-32 they state that with MIMO radar you can't separate different ranges, same angle targets, which is just not true. you can. even without FMCW waveform, e.g. coded waveforms.

2. lines 33-34 the authors introduce a parameter called direction of the department without defining it.

3. lines 34-37 need a complete english overhaul. what is the field of target location? 

4. lines 37-44 are poorly written, and terms are not explained, like the fence effect or the two dimensional field.

5. what is smoothing technology in line 54

6. line 82 introduces a term named U-root-MUSIC without defining it.

7. what is multi-objective localization in lines 95-96 - not defined

8. line 96 "automatically matched" what does it mean?

9. what is small snapshots? line 98

10. item 3 lines 98-101 - i can't understand what the author claim. the english is completely off.

11. table 1-

modulus operator and not modulo

set of MxN matrix should be complex space of size MxN

12. line 128 - does d(m-1) means d*(m-1)? or is it just notation? why if \tau independent of m?

13. 130-132 you write it reflects out of the k-th target but then sum all k targets in eq 3

14. eq.3 should include m index also on the respective \tau's

15. eq. 9 there are two Q matrices, one denoted Q_M and one just Q. the matrix Q is not not defined though. Also it is not clear how Q_M is obtained and what is J_M in this context. 

16. Eq. (11) uses a matrix Gamma without defining it.

17. same for eq. 12. root polynomial obtaining is not defined neither do you quote work on this approach

18. The Pseudo noise resampling method is a part of the paper's title, yet, you devote only a small section to it, and again the math there is unclear and using undefined symbols, like for example eqs 29 and 30 are unclear. 

In general: the math part of the paper is unclear, I suggest breaking the derivation to more steps. 

19. it is unclear what the PN approach is actually doing and how it is improving the results over traditional methods. 

Reviewer 2 Report

Summary:

This is a meaningful work.

Target parameter estimation of FMCW-MIMO radar is a research hotspot in this field in recent years. However, how to realize the estimation method with high accuracy and small error is still of great research space and practical significance.

In order to solve the problem of subspace leakage of traditional estimation methods in the case of small samples and low signal-to-noise ratio. In this manuscript, a new joint DOA and range estimation algorithm for the FMCW MIMO radar system to solve the performance degradation of the above algorithms under small samples and SNR. This method improves the accuracy and stability of target estimation.

The present form of the manuscript is well written, although some minor problems need to be addressed.

The paper can be improved in the following aspects:

Detailed comments:

1. I think the description of FMCW and MIMO in the abstract is too redundant. Failure to focus on the proposed new approach and problem-solving process.

2. English needs further revision to avoid colloquial and grammatical errors.

3. In Figure 1, I think it should be explained in detail how either of the virtual arrays is derived from the transmit and receive antenna arrays. Increase the readability of articles.

4. In Equation (7) in Section 3.1, some of the symbolic representations have been detailed in Table 1 and I do not think it is necessary to re-elaborate.

5. In Figure. 7 of Section 4.2.1, when the number of snapshots is greater than 70, what is the reason for the same RMSE performance between the proposed algorithm and other algorithms? A detailed explanation should be given. 

6. This paper proposes a new joint DOA and range estimation algorithm for the FMCW MIMO radar system. In my opinion, Section 4.2.3 should separately describe the influence of SNR and snapshots on the probability of successful detection, just like Section 4.2.1 and Section 4.2.2.

Author Response

Manuscript ID:sensors-1991162
Dear Editor and Reviewer
We would like to thank the editor and reviewer for giving us helpful suggestions which would help us to improve the quality of the paper. Here, we have revised our paper according to the suggestions of the reviews. We mark all the changes in different color in the revised manuscript. Red represents reviewer 1, blue represents reviewer 2. Thanks very much for your attention to our paper. For your guidance, the attachment is the response to the comments in detail.

Round 2

Reviewer 1 Report

Though you tried to correct, the english level of this paper makes in incomprehensible, and it seems you haven't sent it to english-scientific proofing.

The technical language is sloppy: As an example, words like 'techniques' are interchanged with 'technologies' and the Least-Squares method is sometimes called least square and sometimes squares along the paper, another example is in the results sections -- vertical axis of the graphs are unitless. and there's a lot more.

Author Response

Thank you very much for your comments and efforts spent on our manuscript. In this revision of paper, we have tried ourselves best to improve the readability of the paper.

Please see the attachment for details.

Reviewer 2 Report

No more question. The revised manuscript has been received. The part of the manuscript that needs to be modified has been completed. 

Author Response

We would like to express our sincere thanks to you for your thoughtful and insightful comments. Your valuable comments have helped us improve the structure and content of manuscript. Again, we sincerely appreciate your time and effort in reviewing this manuscript.